# Chronic Effects of Apelin on Cardiovascular Regulation and Angiotensin II-Induced Hypertension

**DOI:** 10.3390/ph16040600

**Published:** 2023-04-17

**Authors:** Qi Zhang, Yue Shen, Sayeman Islam Niloy, Stephen T. O’Rourke, Chengwen Sun

**Affiliations:** Department of Pharmaceutical Sciences, North Dakota State University, Fargo, ND 58105, USA

**Keywords:** apelin, APJ receptor, blood pressure, hypertension, angiotensin II

## Abstract

Apelin, by stimulation of APJ receptors, induces transient blood pressure (BP) reduction and positive inotropic effects. APJ receptors share high homology with the Ang II type 1 receptor; thus, apelin was proposed to play a protective role in cardiovascular disease by antagonizing the actions of Ang II. In this regard, apelin and apelin-mimetics are currently being studied in clinical trials. However, the chronic effect of apelin in cardiovascular regulation has not been fully investigated. In the current study, blood pressure (BP) and heart rate (HR) were recorded using a telemetry implantation approach in conscious rats, before and during chronic subcutaneous infusion of apelin-13, using osmotic minipumps. At the end of the recording, the cardiac myocyte morphology was examined using H&E staining, and cardiac fibrosis was evaluated by Sirius Red in each group of rats. The results demonstrated that the chronic infusion of apelin-13 did not change either BP or HR. However, under the same condition, the chronic infusion of Ang II induced significant BP elevation, cardiac hypertrophy, and fibrosis. Co-administration of apelin-13 did not significantly alter the Ang II-induced elevation in BP, changes in cardiac morphology, and fibrosis. Taken together, our experiments showed an unexpected result indicating that the chronic administration of apelin-13 did not alter basal BP, nor did it change Ang II-induced hypertension and cardiac hypertrophy. The findings suggest that an APJ receptor biased agonist could be a better therapeutic alternative for treatment of hypertension.

## 1. Introduction

Hypertension is a major risk factor for cardiovascular events and is associated with cardiac hypertrophy, fibrosis, and vascular remodeling [1,2]. Renin–angiotensin–aldosterone system (RAAS) plays a very important role in the development of hypertension. By targeting RAAS, angiotensin-converting enzyme (ACE) inhibitors, angiotensin type 1 (AT1) receptor antagonists, and aldosterone receptor blockers have been used successfully in the treatment of hypertension [3]. However, about 15% of hypertensive patients develop drug-resistant hypertension [4]. Therefore, it is urgent to identify a novel therapeutic target to control hypertension and to prevent its comorbidities.

Apelin is an endogenous ligand of APJ receptors [5], a type of G protein-coupled receptor expressed in the cardiovascular system. APJ receptors share high sequence identity with the angiotensin II type 1 (AT1) receptor, ranging from 40% to 50% in the hydrophobic transmembrane regions [6]. However, angiotensin II (Ang II) does not bind to APJ receptors [7]. Thus, it was proposed that apelin could antagonize actions of Ang II on AT1 receptors in the renin–angiotensin system [8]. As such, apelin has been implicated in the pathogenesis of numerous diseases [9], such as obesity, diabetes, hypertension, cardiac hypertrophy, and heart failure. Therefore, it is important to investigate the interaction between apelin/APJ receptors and Ang II/AT1 receptor signaling pathways in cardiovascular regulation.

The apelin gene (APLN) encodes a 77 amino acids preproapelin that is cleaved in C-terminal to several fragments including: apelin-36, apelin-17, and apelin-13 [5]. Apelin-13 is the most active apelin due to its high binding affinity to APJ receptors. The pyroglutamyl form of apelin-13, [pyr^l^]-apelin-13, is resistant to enzymatic cleavage; therefore, it has a longer half-life as compared to other isoforms [10]. Previous studies have demonstrated that intravenous administration of apelin-13 induces a transient depressor effect on blood pressure (BP) and a positive inotropic effect on the heart [11]. It is believed that apelin plays a protective role in cardiovascular diseases, such as heart failure and hypertension. As such, more than fifty clinical trials are being performed in humans (https://www.clinicaltrials.gov/ (accessed on 16 April 2023) to study the possibility that apelin could be used for the treatment of cardiovascular diseases. However, the chronic effect of apelin on cardiovascular function has not been fully studied.

In this regard, the aims of the present study were: (1) To assess the chronic effect of apelin on BP and heart rate (HR); (2) To study the possible interaction between apelin/APJ receptor and Ang II/AT1 receptor signaling pathways in the chronic regulation of cardiovascular function.

## 2. Results

### 2.1. Effect of Chronic Subcutaneous Infusion of Apelin on BP, HR, and Cardiac Morphology

Previous studies have demonstrated that intravenous administration of apelin induces a transient decrease in blood pressure [11,12]. However, the chronic effect of apelin on the cardiovascular system is not fully clear. In this regard, we recorded BP and HR continuously by the telemetry implantation approach in conscious rats, before and during chronic subcutaneous infusions of [pyr^1^]-apelin-13 (500 µg/kg/day), for 2 weeks, using osmotic minipumps. The results are presented in Figure 1, demonstrating that subcutaneous infusions of apelin did not significantly alter the mean arterial pressure (MAP, 98.6 ± 3.7 mmHg versus 99.4 ± 3.3 mmHg, before and after apelin infusion respectively; *n* = 6 rats in each group, *p* > 0.05) (Figure 1A). The HR was not significantly changed by 2-week subcutaneous infusions of apelin-13 (378 ± 7 bpm versus 388 ± 4 bpm, before and after apelin infusion respectively; *n* = 6 rats in each group, *p* > 0.05) (Figure 1B). Subcutaneous infusions of saline had no effect on either MAP or HR (Figure 1A,B). Moreover, chronic subcutaneous infusions of apelin did not significantly alter the systolic arterial pressure (SAP, 123.6 ± 1.6 mmHg versus 118.6 ± 3.9 mmHg, before and after infusion of apelin-13 respectively; *n* = 6 rats in each group, *p* > 0.05) (Figure 1C), diastolic arterial pressure (DAP, 79.3 ± 0.4 mmHg versus 81.0 ± 3.5, before and after infusion of apelin respectively; *n* = 6 rats, *p* > 0.05) (Figure 1D), and pulse arterial pressure (PAP, 44.2 ± 1.4 mmHg versus 37.6 ± 3.7 mmHg, before and after infusion of apelin respectively; *n* = 6 rats in each group, *p* > 0.05) (Figure 1E). Two-week subcutaneous infusions of saline did not significantly change the SAP, DAP, and PAP (Figure 1C–E). Taken together, the results demonstrate that chronic subcutaneous infusions of apelin failed to alter BP and HR.

Previous studies have demonstrated that apelin regulates cardiomyocyte hypertrophy [13]. Thus, we also studied the chronic effect of apelin on cardiac morphology by measurement of the heart weight versus body weight ratio (HW/BW) and the size of cardiac myocyte in the heart sections of rats that received subcutaneous infusions of [pyr^1^]-apelin-13 (500 µg/kg/day) or saline for 2 weeks using osmotic minipumps. The results are presented in Figure 2, indicating that chronic treatment with apelin-13 did not significantly alter the size of cardiac myocytes as compared with the normal saline treatment. HW/BW was also comparable between these two groups of rats.

### 2.2. Chronic Effect of Apelin and Ang II on BP and HR

To rule out the possibility that the lack of effect of apelin in the regulation of BP and HR in the above experiments (Figure 1) was due to a deficit in the technique used, we performed the same procedure to determine the effect of Ang II, a well-known vasoactive peptide, as a positive control on BP and HR regulation in SD rats. The results are presented in Figure 3, demonstrating that chronic subcutaneous infusions of Ang II (150 µg/kg/day) significantly increased MAP, starting 3 days after initiating the infusion, reaching a peak after 6 days, and lasting for 2 weeks during the infusion. The data presented in Figure 3C–E indicate that chronic subcutaneous infusions of Ang II significantly elevated systolic arterial pressure (SAP, 114.3 ± 3.6 mmHg vs. 168.1 ± 6.2 mmHg, before and after Ang II infusion; *n* = 6 rats in each group, *p* < 0.01), diastolic arterial pressure (DAP, 71.2 ± 2.8 mmHg vs. 111.7 ± 4.3 mmHg, before and after Ang II infusion; *n* = 6 rats in each group, *p* < 0.01), and pulse arterial pressure (PAP, 43.5 ± 2.2 mmHg vs. 75.8 ± 3.0 mmHg, before and after Ang II infusion; *n* = 6 rats in each group, *p* < 0.01). However, the HR was not altered by Ang II infusion (Figure 3B). Chronic subcutaneous infusion of saline altered neither MAP nor HR in these rats.

It has been reported that APJ receptors share a 31% amino acid sequence identity with AT1 receptors [6]. Therefore, we examined the possibility that apelin could interact with Ang II/AT1 receptors by detecting the chronic effect of apelin on the Ang II-induced pressor response. The results from this experiment are presented in Figure 3, demonstrating that subcutaneous infusions of apelin-13 did not significantly alter Ang II-induced increases in MAP, SAP, DAP, and PAP. Chronic subcutaneous infusions of apelin-13 plus Ang II did not significantly alter HR in these rats. Together, the results from this experiment suggest that, in contrast to apelin, the chronic subcutaneous infusion of Ang II induces a dramatic increase in BP and that apelin has no significant effect on the Ang II-induced pressor response. Chronic administrations of Ang II or apelin had no significant effect on HR.

### 2.3. Chronic Effect of Apelin and Ang II on Cardiac Morphology

To examine the possibility of an interaction between apelin/APJ receptor and Ang II/AT1 receptor signaling pathways in the heart, we examined the cardiac morphology and the pro-hypertrophic gene expression in the heart of rats that received subcutaneous infusions of saline, Ang II (150 µg/kg/day), and Ang II plus [pyr^1^]-apelin-13 (500 µg/kg/day). The results are presented in Figure 4, demonstrating that the size of cardiac myocytes (Figure 4A,B) was significantly increased in heart sections of rats that received chronic subcutaneous infusions of Ang II as compared with saline control. Furthermore, the ratio of heart weight versus body weight (HW/BW) was also significantly increased by Ang II treatment (Figure 4C), as compared with saline control. The mRNA expression of atrial natriuretic factor (ANF) and beta-myosin heavy chain (β-MHC) in the heart tissue was elevated by chronic Ang II administration, as compared with saline (Figure 4D,E). More interestingly, apelin treatment did not significantly alter Ang II-induced increases in myocyte size, HW/BW, and mRNA expression of ANF and β-MHC. Taken together, the results from this experiment indicate that the chronic treatment with Ang II results in cardiac hypertrophy and that apelin has no significant effect on the Ang II-induced hypertrophy in the heart.

### 2.4. Chronic Effect of Apelin and Ang II on Cardiac Fibrosis

It is well known that hypertension may induce cardiac hypertrophy, fibrosis, and heart failure. Therefore, we also examined the chronic effect of apelin on Ang II-induced cardiac fibrosis. The fibrosis was measured using Sirius Red staining in heart sections of rats that received subcutaneous infusions of normal saline (control), [pyr^1^]-apelin-13 (500 µg/kg/day), Ang II (150 µg/kg/day), or Ang II plus [pyr^1^]-apelin-13 for 2 weeks using osmotic minipumps. The results are presented in Figure 5, indicating that treatment with Ang II dramatically increased the formation of collagen fibers in the heart and that chronic treatment with apelin-13 did not significantly alter Ang II-induced cardiac fibrosis as compared with normal saline treatment.

## 3. Discussion

In this study, we investigated the chronic effects of apelin-13 on cardiovascular regulation in normotensive rats. The major findings of this study are that chronic administration of apelin-13 has no effect on blood pressure, cardiac morphology, and Ang II-induced hypertension. This conclusion is supported by the following lines of evidence: (1) Chronic subcutaneous infusion of apelin-13 did not significantly alter blood pressure, heart rate, and cardiac morphology; (2) In contrast, chronic subcutaneous infusion of Ang II induced a dramatic elevation in blood pressure, severe cardiac hypertrophy, and fibrosis; and (3) Co-treatment with apelin-13 did not significantly alter Ang II-induced blood pressure elevation, cardiac hypertrophy, and cardiac fibrosis. Collectively, unexpected results from this study demonstrate that the chronic administration of apelin did not reduce either the blood pressure or Ang II-induced hypertension, cardiac hypertrophy, and cardiac fibrosis. However, several concerns from this study should be addressed.

Previous studies have demonstrated that intravenous bolus injection of apelin induces a transient decrease in blood pressure [11,12]. The first concern is why chronic administration of apelin did not induce a depressor response in blood pressure. One possible reason could be that long-term administration of apelin induces APJ receptor desensitization in arteries, leading to a reduced response to the APJ agonist, apelin. This hypothesis is supported by a recent important discovery that APJ receptor stimulation leads to β-arrestin-mediated receptor internalization in addition to the G protein-dependent signaling pathway [14,15,16]. APJ receptor internalization could cause desensitization of the receptor to apelin. Therefore, it is crucial to design APJ receptor biased agonists in order to stimulate the APJ-mediated signaling pathways selectively: G protein-dependent versus β-arrestin recruitment-dependent. Research on APJ receptor biased agonists is under intense investigation currently [17,18,19,20]. To examine this hypothesis, our research group is currently designing and evaluating the efficacy of novel APJ receptor biased agonists [18]. It will be interesting to test the chronic effect of APJ receptor biased agonists on regulation of the cardiovascular system in future studies.

The results from this study indicate that the chronic administration of apelin does not significantly improve Ang II-induced elevation in blood pressure and cardiac remodeling, indicating that apelin does not interact with AT1 receptors, although APJ receptors share high sequence similarity with AT1 receptors [6]. However, this result is not consistent with previous studies showing that the chronic administration of apelin is effective in treating preeclampsia, pulmonary arterial hypertension, and myocardial infarction-associated cardiac remodeling [21,22,23]. One possible explanation of this controversy could be due to the hypoxic conditions in those disorders. For example, in the pathophysiology of preeclampsia it is well known that reduced placental perfusion and inadequate cytotrophoblast invasion may cause severe placental hypoxia, which alters gene expression in placental blood vessels and contributes to the development of hypertension [24]. A large number of studies have demonstrated that hypoxia also contributes to the pathogenesis of pulmonary arterial hypertension and myocardial infarction-induced cardiac remodeling [25,26]. More interestingly, recent studies have shown that gene expression of both apelin and APJ receptors is altered by hypoxia, possibly through stimulation of HIF-1α [25]. Taken together, the apelin/APJ-targeting therapy may be more effective for cardiovascular disorders with hypoxic conditions.

It should be noted that this study only examined the chronic effect of apelin by chronic subcutaneous infusion. The apelin in the bloodstream could be degraded by its metabolizing enzymes. Therefore, it is important to evaluate the chronic effect of longer half-life APJ receptor agonists or a biased agonist on cardiovascular regulation, which will be the focus of our future study.

In summary, this study observed the unexpected result that the chronic administration of apelin does not alter blood pressure and heart rate. In contrast, the subcutaneous infusion of Ang II significantly elevates blood pressure, inducing severe cardiac hypertrophy and fibrosis. Co-administration of apelin does not alter Ang II-induced elevation in blood pressure, cardiac hypertrophy, and fibrosis. These results suggest that APJ receptor biased agonists could be a better alternative to target the apelin/APJ receptor system in the treatment of hypertension. Apelin therapy may be more effective for hypoxia-related cardiovascular disorders.

## 4. Material and Methods

### 4.1. Animals and Materials

Male Sprague Dawley (SD) rats (weighting 250–320 g, purchased from Charles River, Wilmington, MA, USA) were used in this study. The rats were housed under controlled conditions with a 12-h light/dark cycle. Rat chow and water were provided ad libitum. All protocols were approved by the North Dakota State University Institutional Animal Care and Use Committee (IACUC) and executed in accordance with the Guide for the Care and Use of Laboratory Animals (National Institutes of Health, Bethesda, MD, USA). In this study, the osmotic minipumps were obtained from ALZET.com. Saline and PBS buffer were purchased from Thermo Fisher, Waltham, MA, USA. Ang II, Apelin-13, and other agents were purchased from Millipore Sigma, St. Louis, MO, USA.

### 4.2. Chronic Recording of BP and HR

Chronic BP and HR were recorded using radiotelemetry under conscious conditions, as described in our previous publication [27]. Briefly, under isoflurane anesthesia, a telemetry BP transducer probe (model TA11PA-C40) was implanted intraabdominally. The transducer probe consists of a transmitter and a fluid-filled catheter. The catheter was inserted into the aorta rostral to the iliac bifurcation, and glued into place with medical adhesive. The sterilized transducer was sutured to the abdominal muscles during the closure of the muscle incision. All surgical procedures were performed under aseptic conditions by following all survival surgery requirements including presurgical procedures, during surgery monitoring, and postsurgical care. Wireless telemetry BP signals were received by a device placed underneath the rat cage, which was connected to a computer equipped with the Dataquest IV system (Data Sciences International, St. Paul, MN, USA). The BP and HR were recorded in the rats under free-moving and conscious condition. Continuous recordings were started five days after the probe implantation to allow the rats to fully recover from the surgery.

### 4.3. Chronic Administration of Apelin

To study the chronic effect of apelin on BP and Ang II-induced hypertension, apelin and/or Ang II were chronically infused subcutaneously via a minipump implanted as described in our previous publication [28]. After recording basal BP for 3 days using telemetry, the chronic infusion minipumps (model 2004, Alzet) were implanted subcutaneously on the back between the shoulder blades of the rats under anesthesia by inhalation of isoflurane (3%). The rats were divided into four groups which received subcutaneous infusion of saline (0.9% NaCl), [pyr^1^]-apelin-13 (500 µg/kg/day), Ang II (150 µg/kg/day), or Ang II plus apelin-13, respectively. After implantation of the infusion pump, BP and HR were recorded daily for an additional 2 weeks. At the end of experiments, the heart and arteries were collected for histological analysis.

### 4.4. Heart Histology and mRNA Detection

To examine the effect of apelin and Ang II on cardiac myocyte morphology, hearts were collected after 2-week BP and HR recording in those four groups of rats as described above. The isolated hearts were washed with pre-cooled saline solution. After weighing, the hearts were transversely sectioned. Some heart sections were fixed in 4% formalin/PBS. Heart sections were prepared and stained with hematoxylin and eosin (H&E). Cardiac myocyte morphology was visualized with a microscope (Olympus, Singapore). The cross-sectional area of single myocytes was measured with Infinity Capture and Analysis Software to evaluate the size of the cardiac myocytes.

To evaluate the effect of apelin and Ang II on cardiac hypertrophy, the expression of hypertrophy-related genes was examined using real-time PCR. The isolated hearts from those four groups of rats were transversely sectioned immediately. Some heart sections were quickly frozen in liquid nitrogen for mRNA detection using the real-time PCR technique as described in our previous publication [29]. Briefly, TaqMan probes specific for atrial natriuretic peptide (ANP), or beta myosin heavy chain (β-MHC) were purchased from Applied Biosystem Inc. (Forster City, CA, USA). The PCR primers for ANP: Forward, 5′-CTGCTAGACCACCTGGAGGA-3′; Reverse, 5′-AAGCTGTTGCAGCCTAGTCC-3′. The PCR primers for β-MHC: Forward, 5′-AGAGCAAAAGCAAAGGGTTTC-3′; Reverse, 5′-GTGATGGTACGAGATGGGCTA-3′. Total RNA was extracted from the heart using RNeasy Mini Kit (Qiagen, Valencia, CA, USA). RNA purity and concentration were determined spectrophotometrically. For each RT-PCR reaction, 2 µg of total RNA was converted into cDNA with a reverse transcriptase (Qiagen). The genomic DNA was eliminated by DNase I. The real-time PCR was performed by placing 10 µL of reaction solution into 96-well plates with SYBER Green or TaqMan PCR Master Mix (Applied Biosystems, Waltham, MA, USA). The oligonucleotide primers were added into the reaction solution. The real-time PCR was performed using an Applied Biosystem PRISM 7000 sequence detection system according to the protocol provided by the manufacturer. A comparative cycle of threshold fluorescence (Ct) method was used with 18S rRNA as an internal control. The Ct value for 18S was subtracted from the Ct value for the gene of interest to give a ΔCt for each sample. The ΔCt of the control was then subtracted from each sample to give a ΔCt value for each sample. The final mRNA levels relative to the control were calculated by equations: (1) ΔΔCt = sample ΔCt – Control ΔCt; (2) final mRNA levels = 2^−ΔΔCt^. In each experiment, samples were examined in triplicate.

To detect cardiac fibrosis, the heart sections were stained with Sirius Red to examine fibrosis in the heart as described in our previous publication [29]. Briefly, the heart sections were fixed in 10% formalin-PBS. Several sections of the heart were prepared with 4–5 µm thickness and stained with Sirius Red to examine collagen fibers. The collagen stained with red color were visualized with a microscope (Olympus) and analyzed with ImageJ software.

### 4.5. Statistical Analyses

All data are expressed as means ± SE. One-way ANOVA and Student’s two-tailed *t* test were used for multiple group comparisons and between group comparisons, respectively. Differences were considered significant at *p* < 0.05. Individual probability values are noted in the figure legends.

## Figures and Tables

**Figure 1 pharmaceuticals-16-00600-f001:**
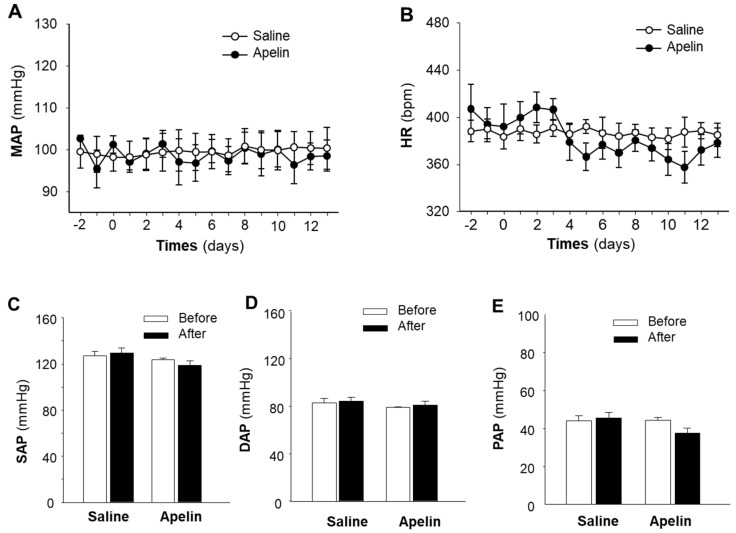
Effect of chronic subcutaneous infusions of apelin on BP and HR. BP and HR were recorded using the telemetry technique in conscious SD rats, before and during subcutaneous infusion of [pyr^1^]-apelin-13 or saline (control) through osmotic minipumps, as described in the Methods. Time courses of changes in MAP (**A**) and HR (**B**) were determined before and during subcutaneous infusions of apelin (filled circle) or saline (empty circle). (**C**–**E**), bar graphs show the systolic arterial pressure (SAP), diastolic arterial pressure (DAP), and pulse arterial pressure (PAP) recorded before (empty bar) and after (filled bar) subcutaneous infusions of apelin-13 or saline. Values are expressed as means ± SE (*n* = 6 for each group).

**Figure 2 pharmaceuticals-16-00600-f002:**
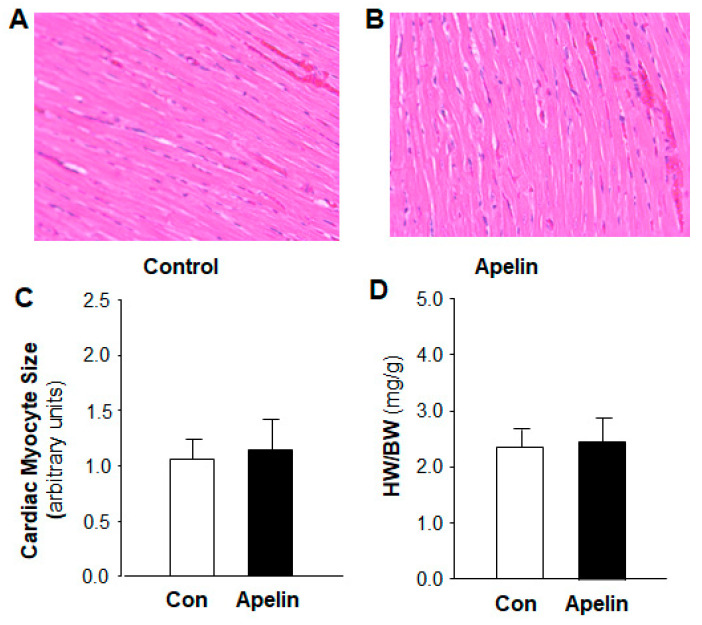
Effect of chronic subcutaneous infusions of apelin on cardiac morphology. The cardiac morphology was examined using H&E staining in the heart sections of rats that received chronic treatment with control (saline) and apelin-13 for two weeks. (**A**,**B**), micrographs (magnification, 20X) showing representative heart sections stained with H&E in rats that received subcutaneous infusion of saline or [pyr^1^]-apelin-13. (**C**), bar graphs summarizing the cardiac myocyte area within the cardiac sections in each group of rats. Data are means ± SE from 6 rats in each group (*p* > 0.5). (**D**), bar graphs showing the ratio of heart weight versus body weight (HW/BW) in rats that received subcutaneous infusion of saline or [pyr1]-apelin-13. Data are means ± SE from 6 rats in each group (*p* > 0.5).

**Figure 3 pharmaceuticals-16-00600-f003:**
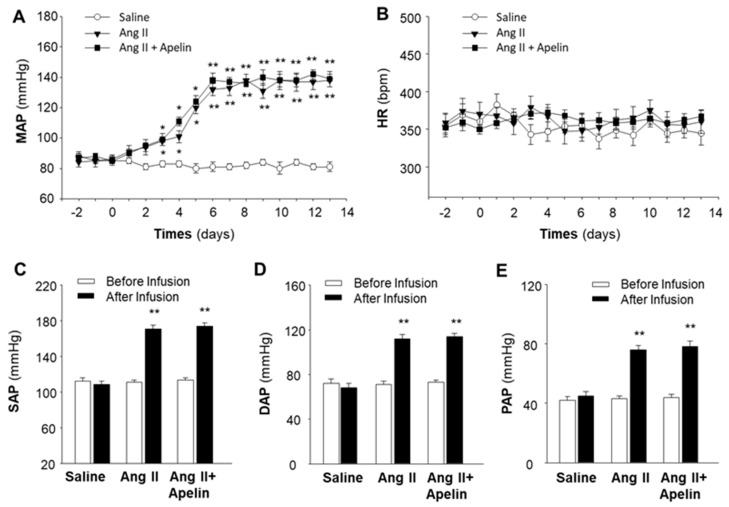
Effect of chronic subcutaneous infusions of apelin and Ang II on blood pressure and heart rate. Blood pressure (BP) and heart rate (HR) were recorded by telemetry implantation in conscious SD rats before and during subcutaneous infusion of saline, Ang II, or Ang II plus [pyr^1^]-apelin-13 through osmotic minipumps, as described in the Methods. Time courses of changes in MAP (**A**) and HR (**B**) were determined before and during subcutaneous infusion of saline (empty circle), Ang II (filled triangle), or Ang II plus apelin-13 (filled square). Values are expressed as means ± SE (*n* = 6 for each group). Significance was set as *p* < 0.05 (*) and *p* < 0.01 (**), comparing with saline control at each corresponding time point. (**C**–**E**), bar graphs showing the systolic arterial pressure (SAP), diastolic arterial pressure (DAP), and pulse arterial pressure (PAP) recorded before (empty bar) and after (filled bar) subcutaneous infusion of saline, Ang II, and Ang II plus apelin-13. Values are expressed as means ± SE (*n* = 6 rats in each group). Significance was set as *p* < 0.01 (**), comparing with the basal value before infusion.

**Figure 4 pharmaceuticals-16-00600-f004:**
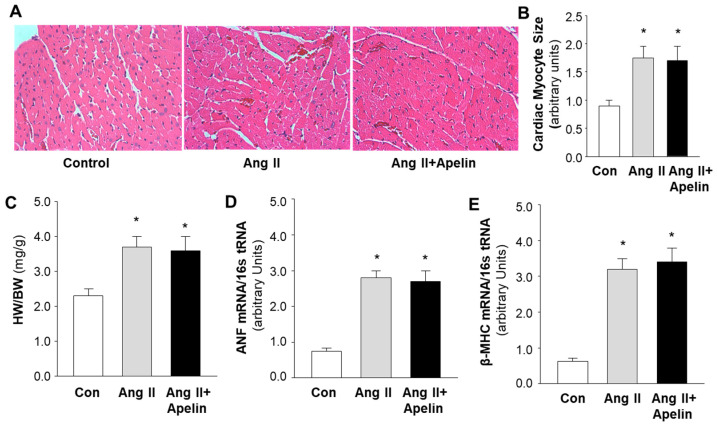
Effect of chronic subcutaneous infusions of apelin and Ang II on cardiac morphology and expression of hypertrophy-related genes. The cardiac histology and gene expression were examined using hematoxylin and eosin (H&E) staining and real-time PCR, respectively, in rats that were treated with saline, Ang II, or Ang II plus apelin, as described in the Methods. (**A**), micrographs (magnification, 20X) showing representative heart sections stained with H&E in rats that received subcutaneous infusions of saline, Ang II, or Ang II plus [pyr^1^]-apelin-13 through osmotic minipumps, as described in the Methods. (**B**), bar graphs summarizing the cardiac myocyte area within the cardiac sections in each group of rats. (**C**–**E**), bar graphs showing the ratio of heart weight versus body weight (HW/BW) (**C**), mRNA levels of atrial natriuretic factor (ANF) (**D**), and beta myosin heavy chain (β-MHC) (**E**), measured using real-time PCR in the heart of rats that received subcutaneous infusion of saline, Ang II, or Ang II plus apelin-13. Data are expressed as means ± SE (*n* = 6 rats in each group). Significance was set as *p* < 0.05 (*), comparing with saline control.

**Figure 5 pharmaceuticals-16-00600-f005:**
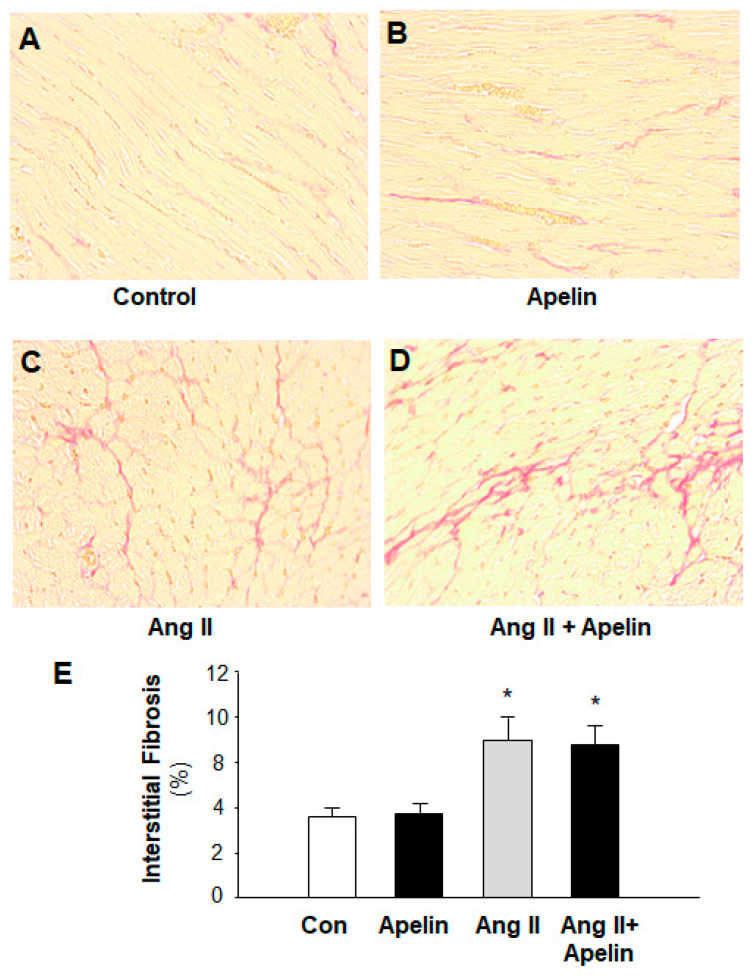
Effect of chronic subcutaneous infusions of apelin and Ang II on cardiac fibrosis. The cardiac fibrosis was measured using Sirius Red in heart sections of rats that received chronic treatments with normal saline (control), apelin, Ang II, and Ang II plus apelin for two weeks. (**A**–**D**), representative microscopic images of the cross-section of the hearts stained with Sirius Red in the rats treated with saline (**A**), [pyr1]-apelin-13 (**B**), Ang II (**C**), and Ang II plus apelin (**D**). (**E**), bar graphs summarizing the cardiac fibrosis in the heart of those rats. Values are expressed as means ± SE (*n* = 6 for each group). Significance was set as *p* < 0.05 (*), comparing with the control.

## Data Availability

Data is contained within the article.

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
