# Peer review of "Chronic Effects of Apelin on Cardiovascular Regulation and Angiotensin II-Induced Hypertension"

_pharmaceuticals, 2023, doi:10.3390/ph16040600_

Round 1

Reviewer 1 Report

The authors address a very interesting topic that is of great interest to the journal's readers. The information provided in the manuscript is enriching, in addition to the fact that the way it is presented is clear and provides relevant information.

Author Response

We thank this reviewer for the comments and recommendations, which have improved this manuscript. As suggested, the references have been updated and checked carefully in the revised manuscript. Thanks again for your suggestion!

Reviewer 2 Report

 I recommend reject this paper

Author Response

We thank this reviewer so much for your time to review our manuscript. Based on your suggestion, we have edited this manuscript accordingly. The grammar and spellings have been checked carefully. Thanks again for your recommendation.

Reviewer 3 Report

Dear Authors,

I am glad to have the opportunity to review your work. This study aimed to evaluate chronic effects of apelin on cardiovascular regulation and angiotensin II-induced hypertension.

The topic is novel and study results indicate that APJ receptor biased agonists should be investigated in the future studies. However, to better understand the study design and the study results, it is necessary that you organize the manuscript according to the standard rules and add some much needed details.

In your paper, you presented Results before the Methodology, which needs to be changed. Also, in the Results section, you write about previous studies – this needs to be moved into discussion. Please organize Results section more clear and concise. In the Methodology, you need to add what is study design and explain study sample. Also, aim needs to be added right after Introduction and it needs to be clear. Abstract needs to be structured and organized in the same order as the paper, especially methodology needs to be corrected in the abstract. Limitations need to be presented before the Conclusion.

Therefore, I recommend major revision of the paper.

Author Response

Reviewer: I am glad to have the opportunity to review your work. This study aimed to evaluate chronic effects of apelin on cardiovascular regulation and angiotensin II-induced hypertension. The topic is novel and study results indicate that APJ receptor biased agonists should be investigated in the future studies. However, to better understand the study design and the study results, it is necessary that you organize the manuscript according to the standard rules and add some much-needed details.

Response: We thank this reviewer for your comments and recommendations, which help to improve this manuscript dramatically. Based on this reviewer’s suggestion, the abstract, methods, results have been updated and re-organized in the revised manuscript. Detailed responses are provided in the following paragraphs:

1) In your paper, you presented Results before the Methodology, which needs to be changed.

Response: Yes, this has been changed. The Methodology has been switched to before Results in the revised manuscript. Authors want to thank this reviewer for the recommendation.

 2) Also, in the Results section, you write about previous studies – this needs to be moved into discussion. Please organize Results section more clear and concise.

Response: Yes, the Results have been re-organized in the revised manuscript. In each section of the Results, the first sentence summarizes why this experiment was performed and how it was performed. This description may help the reader to understand the results collected from each experiment. Thanks for this suggestion.    

3) In the Methodology you need to add what is study design and explain study sample.

Response: Yes, detailed information has now been added to the Methods section. The aim of each experiment has been provided. The group of animals has been added to the revised manuscript. Thanks for this suggestion.

4) Also, aim needs to be added right after Introduction and it needs to be clear.

Response: Yes, the research aims are now provided on Page 2, Lines 65-68. We thank this reviewer for the suggestion. 

5) Abstract needs to be structured and organized in the same order as the paper, especially methodology needs to be corrected in the abstract.

Response: Yes, the methodology is now included in the Abstract (Page 1, lines 11-15). We thank the reviewer for this suggestion.

6) Limitations need to be presented before the Conclusion.

Response: Yes, the limitations are now added before the Conclusion. We thank this reviewer for this recommendation.

Round 2

Reviewer 2 Report

The author did a well-structured work, for its principal aim (studied the protective role in cardiovascular diseases by antagonizing Angiotensin II (Ang II) action) they measured blood pressure and heart rate in rats, as well as, cardiac myocyte morphology was examined using H&E staining, and cardiac fibrosis was evaluated by Sirius Red in each group of rats. However, they did not employ specific tests such as gene expression of both apelin and APJ receptors to justify the controversial results that they got. In conclusion, they demonstrated the potential of APJ receptor-biased agonists as an alternative therapeutic treatment for hypertension.

The main strength of this paper is based on the demonstration of changes in blood pressure and heart rate caused by chronic infusion of Ang II-induced elevated blood pressure and the co-administration of apelin-13. However, I identified that one of the major findings is to prove the elevated blood pressure caused by infusion of Ang II whereas this method is a well-established animal model for hypertension, which means that they did not show something significantly new.

The author made a big effort to improve the quality of its findings whereas, I am still thinking that the research needs better quality and explain why they obtain these controversial results. I am looking forward to knowing what happens with the current study searching an appropriate APJ receptor-biased agonists.

Altogether, the hypothesis that they present is very interesting to find a therapeutic treatment for hypertension using APJ, whereas they did not show these benefits, they did not demonstrate why they got different results than they expected, and it is not well justified why it is happening. Together I recommend rejecting this paper.

Reviewer 3 Report

Dear Authors,

I am glad to have the opportunity to review your work. You have improved the manuscript structure greatly and made your findings more presentable. Also, you have corrected all of the advised suggestions accordingly. Therefore, I suggest to accept the paper.